# Population Response to Air Pollution and the Risk of Coronavirus Disease in Chinese Cities during the Early Pandemic Period

**DOI:** 10.3390/ijerph18052248

**Published:** 2021-02-24

**Authors:** Miryoung Yoon, Jong-Hun Kim, Jisun Sung, Ah-Young Lim, Myung-Jae Hwang, Eun-Hye Kim, Hae-Kwan Cheong

**Affiliations:** Department of Social and Preventive Medicine, Sungkyunkwan University School of Medicine, Suwon 16419, Korea; violet17@skku.edu (M.Y.); kimjh32@skku.edu (J.-H.K.); sung@skku.edu (J.S.); ahyoungjhm@skku.edu (A.-Y.L.); mj6663@skku.edu (M.-J.H.); dmsp14@skku.edu (E.-H.K.)

**Keywords:** environmental epidemiology, air pollution, health behavior, personal protective equipment, mask-wearing

## Abstract

Health behavior is a critical measure in controlling the coronavirus disease 2019 (COVID-19) pandemic. We estimated the effect of health behaviors against air pollution on reducing the risk of COVID-19 during the initial phase of the pandemic. The attack rates of COVID-19 in 159 mainland Chinese cities during the first 2 weeks after the closure of major cities was estimated; air pollution level as a surrogate indicator of the mask-wearing rate. Data on air pollution levels and meteorologic factors 2 weeks prior to the closure were obtained. The attack rate was compared with the level of air pollution using a generalized linear model after adjusting for confounders. When fine particulates (PM_2.5_) and nitrogen dioxide (NO_2_) levels increased by one unit of air quality index (AQI), the infection risk decreased by 0.7% and 3.4%, respectively. When PM_2.5_ levels exceeded 150 (level 4), the infection risk decreased (relative risk, RR = 0.635, 95% confidence interval, CI: 0.442 to 0.912 for level 4; RR = 0.529, 95% CI: 0.337 to 0.830 for level 5; respectively). After controlling for the number of high-speed railway routes, when PM_2.5_ and NO_2_ levels increased by one AQI, relative risk for PM_2.5_ and NO_2_ was 0.990 (95% CI, 0.984 to 0.997) and 0.946 (95% CI, 0.911 to 0.982), respectively, demonstrating a consistently negative association. It is postulated that, during the early phase of the pandemic, the cities with higher air pollution levels may represent the higher practice of mask-wearing to protect from air pollution, which could have acted as a barrier to the transmission of the virus. This study highlights the importance of health behaviors, including mask-wearing for preventing infections.

## 1. Introduction

The first case of coronavirus disease 2019 (COVID-19) was reported in Wuhan, China, in December 2019 [1]. COVID-19, an infectious disease, is caused by severe acute respiratory syndrome coronavirus 2 (SARS-CoV-2) [2]. Its outbreak has progressed to a pandemic over a short period [3]. As of January 2021, there were more than 100 million confirmed cases and more than 2 million deaths due to COVID-19 worldwide [4], and the number of cases are increasing rapidly as the northern hemisphere enters winter.

People are committed to preventing the spread of this disease, for which no vaccines or therapeutics are available thus far. Efforts to prevent the transmission of COVID-19 through respiratory droplets include practices such as wearing masks, washing hands, and wearing personal face films. Countries have been implementing intensive interventions such as social distancing, movement restriction, and stay-at-home orders. The simultaneous involvement of the various measures can significantly prevent the transmission of COVID-19 due to strong interactions between the means. However, due to their simultaneous implementation, it is difficult to assess the effectiveness of each intervention.

Wearing masks is the most widely used means to prevent the transmission of COVID-19. The World Health Organization (WHO) emphasizes wearing a mask as a key step towards preventing the spread of COVID-19 [5]. Its effect can be directly evaluated when the only means of preventing the transmission between humans is a mask. However, it is difficult to evaluate the effectiveness of wearing a mask alone because COVID-19 spreads through person-to-person transmission and measures such as the mask-wearing, lockdowns in the city, movement restrictions, and stay-at-home orders were implemented almost simultaneously. Hence, the effect of wearing masks for preventing the transmission of COVID-19 can be evaluated in the period when public awareness of the mask-wearing in preventing COVID-19 was not high enough. To investigate the effect of wearing masks in a period when people were less aware that COVID-19 is a respiratory infectious disease, there should be a situation where people wear masks for reasons other than that of an infectious disease. For consistency of research results, various population groups in different regions should wear masks for causes other than COVID-19. 

China is a country where air pollution is severe to score the 11th position in air pollution rankings globally, with an annual mean PM_2.5_ concentration of 39.1 μg/m^3^ [6]. During winter, the Chinese government supplies central heating for four months, from approximately November 15 to March 15 of the following year. Due to the increasing air pollution in China, the rate of mask-wearing has increased in the population [7]. 

Few studies have evaluated whether personal hygiene behaviors, such as washing hands or wearing masks, reduce the incidence of certain diseases. A study found that male participants who washed their hands less frequently during the outbreak of influenza A subtype H1N1 virus infection had more flu-like symptoms [8]. However, there is a lack of studies on personal hygiene behaviors and the risks of certain diseases and infections, because there are limited data on the frequency of handwashing, mask-wearing, and the mask-wearing rate in daily life. It is difficult to measure the mask-wearing rate directly over a certain period. Use of a surrogate measure for wearing a mask in the general population may work as an alternative approach. According to a web-based longitudinal survey in China, health information-seeking behavior that retrieved information related to haze on a mobile instant messenger (WeChat) was found to be an essential predictor of the intention of wearing anti-PM_2.5_ masks [7]. Also, during a haze week, mask sales increased 52.35% compared to the previous week in China [9]. When PM_2.5_ increased 10 µg /m^3^, mobile queries for anti-PM_2.5_ mask increased by 3.6 to 8.4% [10]. Zhang and Mu estimated that a 100-point increase in the air quality index (AQI) increases the total purchase of anti-PM_2.5_ facemasks and all facemasks by 70.6% and 50.4%, respectively [11].

Mask-wearing on a day when the air quality was poor while COVID-19 was not recognized as a pandemic by public could have had an unrecognized preventive effect on the transmission of COVID-19. In addition, it was inferred that the effect would have influenced the initial attack rate by city. This study aimed to evaluate the reduction in the risk of infection due to health behaviors by calculating the risk of COVID-19 according to the air pollution level before awareness of the COVID-19 pandemic in major cities in China.

## 2. Materials and Methods

### 2.1. Study Area and Design

Of the 339 cities in mainland China, we excluded cities without air quality measurements, with <10 cases reported during the study period, all cities (17 cities) in the Hubei province, including Wuhan, where the first case of COVID-19 was reported. Finally, 159 cities were analyzed as study subjects. While calculating the number of patients in each city, we excluded patients from other regions (Figure 1).

Pneumonia of unknown etiology was first reported by the WHO on 31 December 2019 [12]. On 9 January 2020, the WHO confirmed that a novel coronavirus had caused the disease. On 14 January 2020, based on experience with respiratory pathogens, it was reported that the 41 confirmed cases in China were likely to be caused by human-to-human transmission [13]. On 23 January 2020, to control the spread of COVID-19, the Chinese government locked down Wuhan city and soon expanded the lockdown to 12 cities in Hubei province [14]. To estimate the effect of mask-wearing due to air pollution on preventing the spread of COVID-19 before the population seriously recognized COVID-19 as an infectious disease, the attack rate of COVID-19 in each city was calculated based on the exposure before 23 January 2020. The average incubation period of cases in Wuhan in the initial epidemic of COVID-19 was 6.4 days (95% confidence interval (CI), 5.6 days to 7.7 days) [15]. The average difference between the onset and reported date was 7.3 days (95% CI, 5.0 days to 9.6 days) [16]. Initially, we considered the onset date till 23 January i.e., before COVID-19 was recognized. However, we used the reported date till 5 February to estimate the onset date because all Chinese cities, excluding the Hubei province, reported cases from 21 January 2020, and the onset date was unknown. Since the estimated onset date of the reported cases on 21 January was 8 January, air quality measurement data were used from 8 January 8 to 23 January (Figure 2). We set the exposure period to 13 days, taking into account an incubation period of 6 days and a lag time between the onset and reported date of 7 days.

### 2.2. Data Collection

We used the reported case data from 21 January to 5 February 2020, provided by doctor Dingxiang’s team (DXY) (cited 5 February 2020) [17]. Public data were obtained from the National Health Commission, provincial health bureau, provincial government, and Hong Kong, Macau, and Taiwan official channels. The dataset was based on Beijing Standard Time and it provided data on confirmed, death, and cure cases by region. The weather data including temperature data by time and measurement point were provided by Free meteo [18]. Air quality data, including data on daily and average particulate matter with a PM_2.5_, nitrogen dioxide (NO_2_), and AQI, were provided by the World Air Quality Project [19]. Population data were obtained from the Sixth National Population Census of the People’s Republic of China and the National Bureau of Statistics of China [20]. 

### 2.3. Data Analysis

In China, PM_2.5_ concentration is classified into six levels according to PM_2.5_ AQI to rapidly determine the level of air pollution and the resulting health effects. The six levels were good (0–50 AQI, 0–12.0 μg/m^3^), moderate (51–100 AQI, 12.1–35.4 μg/m^3^), unhealthy for sensitive groups (101–150 AQI, 35.5–55.4 μg/m^3^), unhealthy (151–200 AQI, 55.5–150.4 μg/m^3^), very unhealthy (201–300 AQI, 150.5–250.4 μg/m^3^), and hazardous (≥301 AQI, ≥250.5 μg/m^3^) [6]. We assumed that the population wore masks on days when the AQI exceeded 150, the level that affects the general population’s health. The log(attack rate) slope for the days when PM_2.5_ exceeded 150 AQI adjusted for active hours (from 7 a.m. to 7 p.m.) and the maximum temperature was calculated. According to a study on the spatial infectious disease dynamics of the outbreak of COVID-19 in China, the spread of this disease in China was spatially dependent [21,22]. The existence of early confirmed cases in the other regions was assumed to be due to connection with Wuhan’s by express trains. Hence, COVID-19 spread to other regions via people travelling to those regions [21,22]. We made inferences to avoid overestimating the attack rates in areas adjacent to Hubei province and areas with many busy traffics. The way to infer this was to consider the number of high-speed railway routes in each region, that is, by measuring China’s connectivity, rather than the distance from Wuhan. Thus, the number of high-speed railway routes was entered into the model as a factor variable.

### 2.4. Statistical Analysis

The generalized additive model (GAM) is flexible and practical for performing nonlinear regression analyses in time-series studies of the health effects of air pollution [23]. In this study, the relationship between the COVID-19 attack rate and PM_2.5_ and NO_2_ levels was calculated using a Poisson-distributed GAM model.
(1)$$Y_{c}\;\sim \;\mathrm{Poisson}\left(\mu c\right)\;\log\;\mu c\;=\;\alpha \;+\beta \left({\mathrm{PM}}_{2.5\_m}/{\mathrm{NO}}_{2\_m}\right)\;+\;s\left({\mathrm{temperature}}_{m},\;3\right)\;+\;\mathrm{factor}(\mathrm{hsr})$$
where Y_c_ denotes the attack rate in each city from 21 January to 5 February; β denotes the log of relative attack rate associated with a 1-AQI increase in a PM_2.5_m_ or average of NO_2_m_; PM_2.5_m_/NO_2_m_ is the average PM_2.5_ or NO_2_ levels for a specific period by city; *s* (temperature_m_, 3) is a smooth function of daily average temperature for 24 h and daily average temperature for daytime (from 7 a.m. to 7 p.m.); and factor (hsr) is an indicator variable for high-speed railway routes.

Even if the PM_2.5_ concentration increases, the proportion of the mask-wearing population does not increase linearly. The population will wear a mask when the air quality reaches a certain level at which it begins to feel it is a health threat. Hence, the decision to wear a mask depends on the level of subjective health threat or the AQI. It is necessary to measure the attack rate threshold (local extremum point) according to the change in PM_2.5_ concentrations to determine the preventive effect of wearing a mask. A change in slope test (daviest.test in R(The R Foundation for Statistical Computing, Vienna, Austria)) was performed using general linear model (GLM) analysis of attack rate and PM_2.5_ AQI to measure the threshold. davies.test tests for a nonzero slope difference parameter test of the segmented relationship. The null hypothesis is H_0_: β = 0, where β is the difference-in-slopes [24]. We used the segmented package in R Studio to work on this test. In general, the population judges the severity of air pollution according to the PM_2.5_ level manifested by AQI rather than the PM_2.5_ concentration. We calculated the attack rate by the AQI level to measure the relationship between the attack rate and average PM_2.5_ level by each city for a specific period in this study. An AQI level of 1, 2, 3, 4, 5, and 6, is considered good, moderate, unhealthy for sensitive groups, unhealthy, very unhealthy, and hazardous. The PM_2.5_ AQI level was used as a categorical and continuous variable in the GLM. The significance level was set at 0.05. R version 4.02 and SAS version 9.1 (SAS Institute, Inc., Cary, NC, USA) were used for all statistical analyses.

## 3. Results

The median of cumulative number of cases in a city was 24, with the lowest at 10 and the highest at 376, and the median attack rate was 0.56/100,000 (interquartile range (IQR), 0.58) from 21 January to 5 February in all the 159 Chinese cities. The median PM_2.5_ AQI from 8 January to 23 January 2020, was 146.06 (IQR, 64.69). The median PM_2.5_ level was level 3 (IQR, 1.0). The median number of days of PM_2.5_ levels exceeded 150 AQI and 100 AQI were 8.0 days (IQR, 10.0 days) and 14.0 days (IQR, 7.0 days), respectively. The average temperature in the 159 cities was 4.3 °C (standard deviation (SD), 8.5 °C), and the daytime average temperature from 7 a.m. to 7 p.m. was 5.5 °C (SD, 8.3 °C) from 8 January to 23 January 2020 (Table 1).

During the study period, the city with the highest attack rate of COVID-19 was Xinyu, Jiangxi province (6.06/100,000 persons). In Xinyu, the average PM_2.5_ AQI was 112.9 at the 25.2 percentile, the population density was 360 persons/km^2^, the straight distance from Wuhan is 316 km, and there is one high-speed railway route. The city with the lowest attack rate was Heze, Shandong province (0.14/100,000 persons). The average PM_2.5_ AQI in Heze was 211.9 at the 93.7 percentile. The population density was 680 persons/km^2^, the straight distance from Wuhan was 530 km, and there were no high-speed railway routes. The city with the highest PM_2.5_ AQI was Anyang, Henan province, with an average PM_2.5_ AQI of 244.31. In Anyang, the attack rate was 0.68/100,000 persons at 62.3 percentile, and the population density is 700 persons/km^2^, the straight distance from Wuhan is 614 km, and there are no high-speed railway routes. The city with the lowest PM_2.5_ AQI was Sanya, Hainan province, with an average PM_2.5_ AQI of 56.44. In Sanya, the attack rate was 3.21/100,000 persons at 98.1 percentile, the population density is 360 persons/km^2^, the straight distance from Wuhan was 1456 km, and there are no high-speed railway routes (Appendix A).

Pearson correlation between the attack rate and each measured variable during the study period showed a statistically significant positive correlation with the average temperature and population density, and a statistically significant negative correlation with the average PM_2.5_, PM_10_, and NO_2_ levels (Figure 3). In addition, since the correlation between the attack rate and air quality (PM_2.5_, PM_10_, NO_2_) had a stronger negative correlation than that between the attack rate and the distance from Wuhan, the initial attack rate was not due to the spatial effect of Wuhan. It can be inferred that health behavior was more affected by air quality (Figure 3). 

COVID-19 spreads by person-to-person transmission. It is affected by factors such as the natural environment (temperature, humidity, air pollution), geographical factors (altitude, distance from Wuhan), and population and social factors (population, population density). Average temperatures by city, maximum temperature, minimum temperature, daytime average temperature, daytime maximum temperature, daytime minimum temperature, relative humidity, air pollution (PM_10_, O_3_, CO, SO_2_), topographic characteristics (altitude, distance from Wuhan), demographic characteristics (population, population density), and attack rate were not statistically significant as per the GAM model (Figure 4). However, the log(relative rate of attack) for PM_2.5_ and NO_2_ were β = −0.007 (95% CI, −0.011 to −0.003) and β = −0.034 (95% CI, −0.059 to −0.009), respectively, showing a statistically significant negative relationship (Figure 4). The log(relative rate of attack) for the days on which PM_2.5_ exceeded AQI 150 was −0.052 (CI, −0.100 to −0.004) and showed a negative β, which was statistically significant. The activity time and maximum temperature were controlled (from 7 a.m. to 7 p.m.) (Figure 5).

There were no cases in the PM_2.5_ AQI level of 1 (0–50 AQI, 0–12.0 μg/m^3^) and PM_2.5_ AQI level of 6 (≥301 AQI, ≥250.5 μg/m^3^) during the study period. The relative risk of COVID-19 infection was not significant in cities with AQI level of 3 compared to the cities with AQI level 2. However, it was significant in the cities with AQI level of 4 and 5 (RR = 0.635, 95% CI: 0.442 to 0.912 for level 4; RR = 0.529, 95% CI: 0.337 to 0.830 for level 5; respectively) (Figure 6).

We estimated the risk of COVID-19 after entering the number of high-speed railway routes as a confounder into the model. The relative risk of COVID-19 for PM_2.5_ and NO_2_ was 0.994 (95% CI: 0.988 to 1.000) and 0.975 (95% CI, 0.946 to 1.006), respectively. However, when high-speed railway routes were adjusted for the model, relative risk for PM_2.5_ and NO_2_ was 0.990 (95% CI, 0.984 to 0.997) and 0.946 (95% CI, 0.911 to 0.982), respectively (Figure 7).

## 4. Discussion

This study used air pollution as a surrogate indicator of the health behavior of wearing a mask and evaluated the effect of health behavior on reducing the risk of COVID-19 in the initial period of the pandemic. The attack rates by city did not have statistically significant linear or nonlinear relationships with climatic factors (temperature and humidity), topographic factors (altitude, distance from Wuhan), and population factors (population, population density). However, level of air pollution (PM_2.5,_ NO_2_) in the previous two weeks had a significant negative nonlinear relationship with the initial development of COVID-19 in the Chinese cities.

Most of the studies on air pollution and COVID-19 have focused on the relationship between air pollution and COVID-19. Wu et al. reported that when PM_2.5_ concentration increased by 1 μg/m^3^, COVID-19 mortality increased by 8% [25]. Liang et al. reported that the COVID-19 fatality and mortality rates increased by 7.1% and 11.32%, respectively, as the NO_2_ concentrations increased by 4.6 ppb [26]. Yao et al. reported that the case fatality rate after adjusting for temperature, relative humidity, SO_2_, NO_2_, CO, and O_3_ had positive correlations with all Lag0–Lag5 day concentrations of PM_2.5_ and PM_10_ [27]. Zhang et al. reported positive correlations between PM_2.5_, PM_10_, and NO_2_ exposure with Lag14 and 28 days and the average case fatality rate (CFR) of China outside Hubei province [28]. There has also been a study on air quality improvement and health benefits resulting from COVID-19, such as that regarding an intervention to deter COVID-19 that improved the air quality, bringing health benefits [29] and improved air quality due to COVID-10-related urban closure [30]. COVID-19 research using air pollution data has been mainly focused on the impact of air pollution on COVID-19 mortality and fatality rates or the reduction of air pollution due to the intervention effect to prevent COVID-19 and the resulting health benefits.

This study attempted to evaluate the risk of infection using the air pollution level as a surrogate indicator of the mask-wearing rate in the early epidemic period when behavioral changes for the protection of the COVID-19 were not widely practiced. The high frequency of PM_10_ concentration peak (exceeding 50 µg/m^3^) accelerated the spread of COVID-19 in an abnormal COVID-19 outbreak in northern Italy. It is suggested that the PM acts as a vehicle of the SARS-CoV-2 [31]. Wearing a mask in people’s daily life is suitable as an effective means of protection [32]. Thus, wearing a mask can reduce the risk of infection of COVID-19 by preventing air pollutants. The small speech droplets of SARS-CoV-2 carriers have the potential to transmit viruses [33]. In this way, COVID-19 is infected through the SARS-CoV-2 in the droplets of patients, and air pollutants accelerate the spread of COVID-19. Hence, wearing a mask is an important role in reducing COVID-19 infection due to preventing droplets and air pollutants simultaneously. Before the COVID-19 pandemic in China, the population wore a mask by health behavior due to air pollution, not to prevent infectious diseases. Thus, it is appropriate to use the air pollution level as a surrogate indicator of the mask-wearing rate due to air pollution.

This study is differentiated from previous studies; it estimated the effect of reducing the transmission of COVID-19 using real-time air pollution data, which show the temporal and spatial characteristics as surrogate indicators for the health behavior of wearing a mask. Personal health behaviors contribute to the prevention of the spread of infectious diseases. As a result of strict adherence to personal hygiene rules such as handwashing in 20 Korean hospitals in 2020, the detection rate of respiratory viral infections decreased by 50% and that of enteroviral infections decreased by 80% compared to the previous year [34]. This study is based on the fact that personal hygiene actions such as mask-wearing or washing hands contributes to prevention of the transmission of all infectious diseases.

COVID-19 is a novel infectious disease and is an emerging infectious disease in all countries and regions. An emerging infectious disease subsides when it reaches herd immunity or when the rate of infection is decreased with decisive intervention. This study excluded COVID-19-related intervention by specifically considering the initial period when the infectivity of COVID-19 was not well-recognized enough to assess the reduction efficiency of unintended interventions through health behavior before social and national interventions. In addition, to avoid selection bias, the study’s internal validity was considered by excluding the number of cases in Hubei province, including Wuhan, cities with <10 cases in during the study period, and inflow cases from other cities. COVID-19 is a human-to-human transmitted infectious disease; therefore, it is generally expected that the attack rate of COVID-19 will be high in a densely populated area. However, in China, the attack rate was not strongly correlated with demographic or topographic factors before its infectious nature was known. However, it is necessary to consider the differences in the geographical factors between Wuhan and other cities, the first outbreak area, since COVID-19 is an infectious disease with a transmission mechanism. We considered the number of high-speed railway routes for each city because it is more suitable to consider the movement through transportation than to consider Wuhan’s distance from each city.

In this study, the initial attack rate of COVID-19 varied by city, suggesting that COVID-19 might have encountered an unrecognized intervention, even in the early phase when the infection was unknown. We estimated that health behaviors might have been an unintended intervention because cities with severe air pollution (PM_2.5_ and NO_2_) tended to have lower attack rates during the initial infection than cities with no severe air pollution. Generally, wearing a mask is the most popular activity to prevent health risks due to air pollution. According to a study using the mask purchase index data of the Alibaba Group’s Taobao.com and Tmall.com in China, a 100-point increase in AQI increased anti-PM_2.5_ face mask sales by 70.6% and all face mask sales by 54.5% [11]. According to a study on the annoyance score against the levels of air pollution (PM_2.5_ and NO_2_) in the study of air pollution exposure distributions within adult urban populations in Europe (EXPOLIS), the air pollution annoyance score due to traffic congestion was highly correlated with the concentrations of outdoor PM_2.5_ and NO_2_ [35]. Based on these studies, this study used air pollution data as a surrogate indicator for wearing a mask.

The results of this study indicate that wearing a mask due to air pollution reduced the risk of COVID-19. Wearing a mask reduced the frequency of detection of respiratory viruses and viral shedding in respiratory droplets and aerosols from COVID-19 patients [36]. This is consistent with the results of previous studies, which reported that wearing a mask has an essential effect on COVID-19 control [36].

The limitation of this study is that data on the mask-wearing rate due to air pollution could not be used to evaluate the effect of the health behavior of wearing a mask on reducing the risk of COVID-19. Except for a limited short-time survey in one area, studies on the mask-wearing rate have not been conducted in the general environment and general population. Hence, we determined that air pollution data are suitable as a surrogate indicator for the mask-wearing rate by referring to the mask purchase index modeling study for air pollution, the annoyance score study according to the air pollution, and the health information seeking behavior study.

## 5. Conclusions

This study evaluated the effect of unintended health behaviors against air pollution on reducing the initial COVID-19 attack rate. It demonstrated the effectiveness of early interventions to prevent transmission in the initial epidemic. The finding provides evidence for the importance of wearing a mask to prevent infection in a situation when vaccines and therapeutics are not available and the population has not attained herd immunity.

## Figures and Tables

**Figure 1 ijerph-18-02248-f001:**
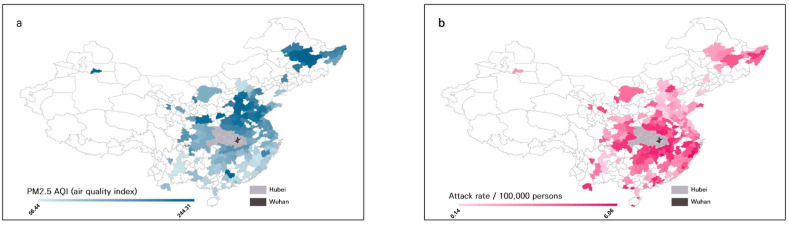
Spatial distribution of air pollution levels and COVID-19 attack rates in 159 cities in mainland China during the early period of the pandemic. (**a**) Average PM_2.5_ air quality index (AQI) mean of 159 cities between 8 January and 23 January 2020. (**b**) Attack rate/100,000 persons on 5 February 2020.

**Figure 2 ijerph-18-02248-f002:**
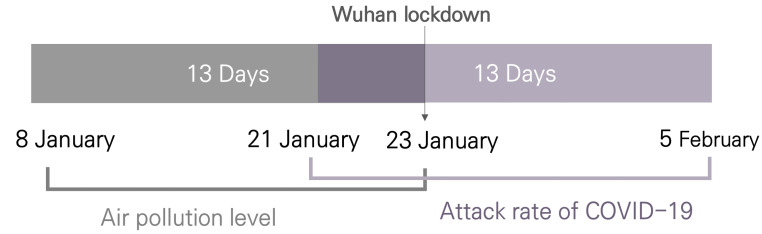
Temporal dimension of the study (Lockdown of Wuhan and other parts of China began on 23 January 2020. Air pollution levels were estimated from 8 January to 23 January 2020. The attack rate of COVID-19 was estimated from 21 January to 5 February).

**Figure 3 ijerph-18-02248-f003:**
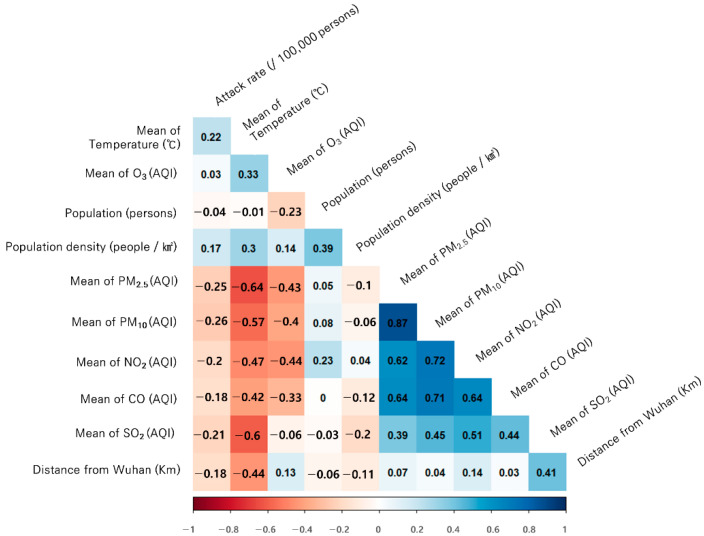
Pearson correlation coefficients for variables and attack rates (The correlation coefficients between attack rates and air quality, topographic characteristics, and demographic characteristics in the study period.).

**Figure 4 ijerph-18-02248-f004:**
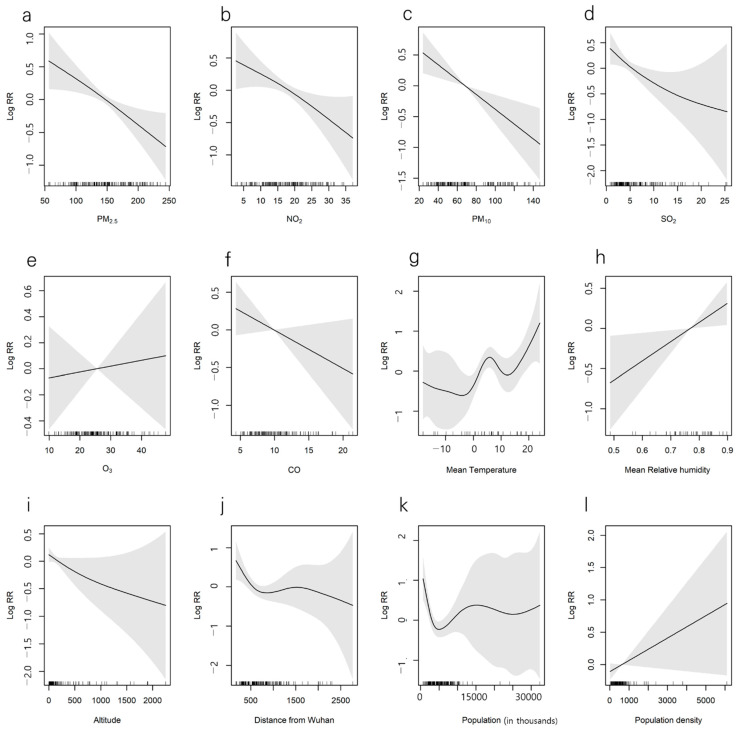
Generalized additive model analysis of the association between attack rates of COVID-19 and meteorologic factors (**g**,**h**), population (**k**,**l**), geographical factors (**i**,**j**), and air quality (**a**–**f**). (Most models did not reveal statistically significant results. However, only models of PM_2.5_ and NO_2_ revealed statistically significant results.).

**Figure 5 ijerph-18-02248-f005:**
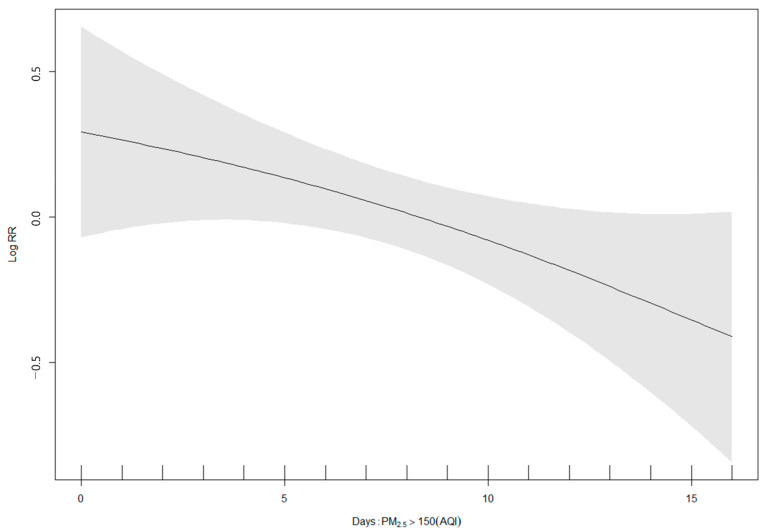
Generalized additive model analysis of the association between attack rates of COVID-19 and the number of days on which AQI exceeded 150 among the 159 Chinese cities during early lockdown period.

**Figure 6 ijerph-18-02248-f006:**
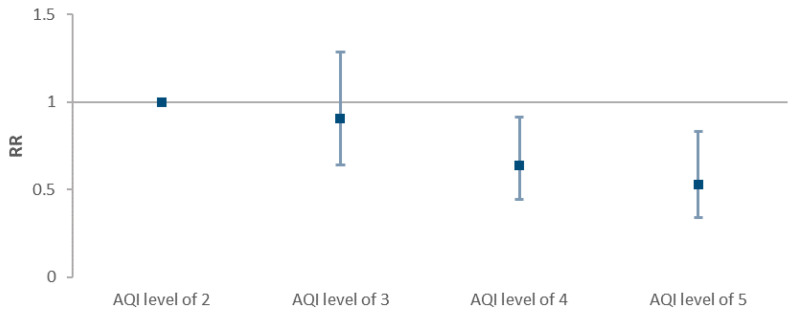
General linear model analysis of the attack rates in 159 Chinese cities and PM_2.5_ AQI levels.

**Figure 7 ijerph-18-02248-f007:**
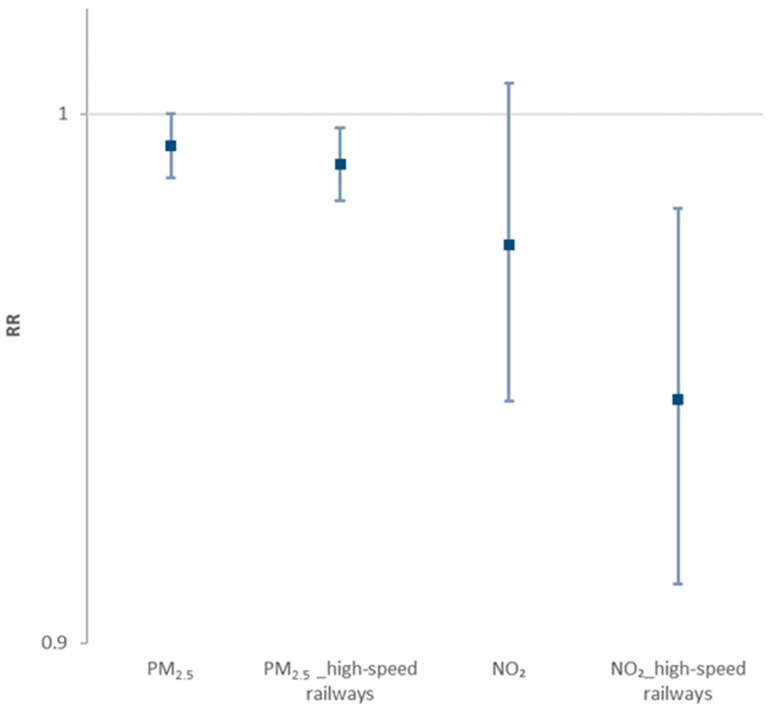
Generalized linear model analysis of the relative risk of COVID-19 in 159 Chinese cities and PM_2.5_ and NO_2_ concentration, with and without adjustment for high-speed railway routes.

**Table 1 ijerph-18-02248-t001:** Descriptive statistics of cumulative cases, attack rate, PM_2.5_, PM_10_, NO_2_, SO_2_, O_3_, temperature, and geographical factors across 159 cities.

Variable	Number of Cities ^9^	Mean	Standard Deviation	Percentile					
				Minimum	25th	Median	75th	Maximum	Inter-Quartile Range
Cumulative cases	159	42.90	57.06	10.00	15.00	24.00	45.00	376.00	30.00
Attack rate/100,000 persons	159	0.79	0.79	0.14	0.36	0.56	0.94	6.06	0.58
PM_2.5_ (AQI) ^1^	159	145.93	41.85	56.44	112.94	146.06	177.63	244.31	64.69
PM_2.5_ AQI level ^2^	159	3.38	0.90	2.00	3.00	3.00	4.00	5.00	1.00
PM_2.5_ ≥ 150 AQI Days ^3^	159	7.52	5.35	0.00	2.00	8.00	12.00	16.00	10.00
PM_2.5_ ≥ 100 AQI Days ^4^	159	12.19	4.28	0.00	9.00	14.00	16.00	16.00	7.00
PM_10_ (AQI) ^5^	154	68.42	28.95	24.19	45.56	59.81	92.50	147.00	46.94
NO_2_ (AQI) ^6^	154	17.65	7.45	2.75	12.44	17.22	22.69	37.06	10.25
SO_2_ (AQI) ^7^	154	6.00	4.70	0.81	2.94	4.16	7.50	25.44	4.56
O_3_ (AQI) ^8^	141	25.54	7.31	9.94	20.56	24.38	29.38	47.81	8.81
Temperature (℃)	159	4.32	8.47	−18.10	0.80	4.05	8.77	24.00	7.96
Day time temperature (7:00–19:00, ℃)	159	5.45	8.25	−16.65	2.45	4.91	9.39	25.79	6.94
Population (1,000,000 persons)	159	5.65	3.95	0.69	3.26	4.81	7.20	32.35	3.93
Population density (persons/km^2^)	158	621.15	672.83	23.00	270.00	508.00	700.00	6100.00	430.00
Distance from Wuhan (km)	159	784.64	449.66	172.00	527.00	705.00	927.00	2768.00	400.00
Number of high-speed railway routes	159	0.49	0.69	0.00	0.00	0.00	1.00	3.00	1.00

^1^ PM_2.5_: particulate matter with diameter 2.5 micrometer or less; AQI, air quality index. ^2^ PM_2.5_ AQI levels: level 1, good (0–50 AQI, 0–12.0 μg/m^3^); level 2, moderate (51–100 AQI, 12.1–35.4 μg/m^3^); level 3, unhealthy for sensitive Groups (101–150 AQI, 35.5–55.4 μg/m^3^); level 4, unhealthy (151–200 AQI, 55.5–150.4 μg/m^3^); level 5, very unhealthy (201–300 AQI, 150.5–250.4 μg/m^3^); and, level 6, hazardous (≥301 AQI, ≥250.5 μg/m^3^). ^3^ PM_2.5_ > 150 AQI days: The number of days PM_2.5_ was greater than 150 AQI during the study period. ^4^ PM_2.5_ ≥ 100 AQI Days: The number of days PM_2.5_ was greater than 100 AQI during the study period. ^5^ PM_10_, particulate matter with diameter 10 micrometer or less. ^6^ NO_2_, nitrogen dioxide. ^7^ SO_2_, sulfur dioxide. ^8^ O_3_, ozone.^9^ Number of cities: Variables with fewer than 159 cities are those with missing values (refer to the appendix).

## Data Availability

Publicly available datasets were analyzed in this study. The data presented in this study are openly available in: (https://ncov.dxy.cn/ncovh5/view/pneumonia (accessed on 24 February 2021)).

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
