# Peer review of "Population Response to Air Pollution and the Risk of Coronavirus Disease in Chinese Cities during the Early Pandemic Period"

_ijerph, 2021, doi:10.3390/ijerph18052248_

Round 1
Reviewer 1 Report
There is a mismatching. The abstract and conclusion present a series of association between air quality index and then the authors conclude speaking about healthy behaviours of people . In the last sentence of abstract and in the conclusion the authors cannot focus only on the healthy behaviours but they must say that "Improvements in Air Quality Index and Reduction in Air pollutants are associated with reduced risk of COVID-19 transmission in a clear way.
In the abstract and in the Results, the authors must say something not only concerning the effects of AQI improvements but also about the negative effect that high concentration of PM and reduction of AQI have on COVID.19 cases. The study should run a model that can assess the impact of reduced air quality and increased PM on the increase OF covid-19 cases.
Bibliography is lacking of the studies published by Setti et al on IJERPH and BMJ Open.
Author Response
Reviewer #1
- There is a mismatching. The abstract and conclusion present a series of association between air quality index and then the authors conclude speaking about healthy behaviours of people. In the last sentence of abstract and in the conclusion the authors cannot focus only on the healthy behaviours but they must say that "Improvements in Air Quality Index and Reduction in Air pollutants are associated with reduced risk of COVID-19 transmission in a clear way.
Response: Thank you for the comment on the consistency of the statement. We have revised the abstract and conclusion based on the findings of this study (L20-23, L374-378)
- In the abstract and in the Results, the authors must say something not only concerning the effects of AQI improvements but also about the negative effect that high concentration of PM and reduction of AQI have on COVID-19 cases.
Response:
Following your suggestion, we have included on the negative association between PM2.5 in conc and AQI, and NO2 show negative association with the COVID-19 risk (L252-270). We also added precedent studies that air pollution spread COVID-19 in the introduction and discussion (L69-78, L295-312).
- The study should run a model that can assess the impact of reduced air quality and increased PM on the increase of covid-19 cases.
Response:
This study's key objective is not the impact of air pollution on COVID-19 but an evaluation of the effect of wearing a mask on the prevent COVID-19 by using air pollution level as a surrogate indicator for wearing a mask. We have conducted both GAM in addition to GLM to demonstrate the dose-response relationship between air pollution level and COVID-19 incidence. It demonstrates that there is a consistently negative relationship between air pollution level and COVID-19 incidence. This relationship was found across the air pollutants, PM10, PM2.5, NO2, and SO2, In addition, We found that the mask worn because of air pollution prevented COVID-19 in China cities. We agree that a model is needed to evaluate the impact of increased air pollution on the increase of COVID-19.
However, Our study region China is different from the situation in Italy, which has been demonstrated in previous study (Setti et al.). In China, where the air pollution level is high, a mask was worn for health behavior caused by air pollution, not to prevent COVID-19.
Cities with high air pollution levels in China are generally large cities or industrial cities, where are population density and people's contact and movement are active. Thus COVID-19 can be expected to spread rapidly. However, cities with high levels of air pollution in China showed to be lower risk of COVID-19 during the initial pandemic (Figure R11).
We decide that a model which is air pollution contributes to the spread of COVID-19 is not suitable for this study. The reason is that the effect of wearing a mask due to air pollution was reflected.
Figure R11. Attack rate and the number of days on which PM2.5 air quality index exceed 150 (see attached file).
- Bibliography is lacking the studies published by Setti et al on IJERPH and BMJ Open.
Response: Thank you for the suggestion and we have added these articles in the revised manuscript and reference list (L299-302, reference 31, 32).
References
- Zhang, J.; Mu, Q. Air pollution and defensive expenditures: evidence from particulate-filtering facemasks. J. Environ. Econ. Manag. 2018, 92, 517-536.
- Rotko, T.; Oglesby, L.; Künzli, N.; Carrer, P.; Nieuwenhuijsen, M.J.; Jantunen, M. Determinants of perceived air pollution annoyance and association between annoyance scores and air pollution (PM2.5, NO2) concentrations in the European EXPOLIS study. Atmos. Environ. 2002, 36(29), 4593-4602
- Liu, T.; He, G.; Lau, A. Avoidance behavior against air pollution: evidence from online search indices for anti-PM 2.5 masks and air filters in Chinese cities. Environ. Econom. Policy Stud. 2018, 20(2), 325-363.
- Zhao, S.; Zhuang, Z.; Ran, J.; Lin, J.; Yang, G.; Yang, L.; He, D. The association between domestic train transportation and novel coronavirus (2019-nCoV) outbreak in China from 2019 to 2020: A data-driven correlational report. Travel Med. Infect. Dis. 2020, 33, 101568.
- Kang, D.; Choi, H.; Kim, J.H.; Choi, J. Spatial epidemic dynamics of the COVID-19 outbreak in China. Int. J. Infect. Dis. 2020, 94, 96-102.
- Copiello, S.; Grillenzoni, C; The spread of 2019-nCoV in China was primarily driven by population density. Comment on “Association between short-term exposure to air pollution and COVID-19 infection: Evidence from China” by Zhu et al.. Sci Total Environ 2020 20, 744,141028
- CoÅŸkun, H; Yıldırım, N; Gündüzc, S. The spread of COVID-19 virus through population density and wind in Turkey cities. Sci. Total Environ. 2021,751,
8.Sun, C.; Kahn, M.E.; Zheng S. Self-protection investment exacerbates air pollution exposure inequality in urban China. Ecol. Econ. 2017, 131, 468-474
- Ban, J.; Zhou, L.; Zhang, Y.; Anderson, G.B.; Li, T. The health policy implications of individual adaptive behavior responses to smog pollution in urban China. Environ. Int.2017, 106, 144–152.
- Zhang, J.; Mu, Q. Air pollution and defensive expenditures: evidence from particulate-filtering facemasks. J. Environ. Econ. Manag. 2018, 92, 517-536.
- Center for disease control and prevention (CDC). Personal Protective Equipment: Questions and Answers. Available online: https://www.cdc.gov/coronavirus/2019-ncov/hcp/respirator-use-faq.html (accessed on 25 January 2021).
Baidu*: Ranks among the top in the world by Alexa.com traffic metrics (cited 2021 Jan 22. https://www.alexa.com/topsites)

Reviewer 2 Report
I write you in regards to manuscript ijerph-1062570 entitled "Population response to air pollution and the risk of coronavirus disease in Chinese cities during the early pandemic period” which you submitted to International Journal of Environmental Research and Public Health.
This study targets the effect of health behaviors against air pollution on reducing the risk of COVID-19 during the initial pandemic. The attack rate was further compared with the level of air pollution using a generalized linear model after adjusting for confounders. The study also evaluated the health behaviors towards using face mask to prevent infections.
This study aims to add knowledge about air pollution and the risk of coronavirus in different areas in China. However, the authors seem to be too ambitious using a rather small dataset (2 weeks) to predict the abovementioned association. I am particularly unsure about the data representativeness for the generalized additive model (GAM) outcome. I also found the links between several air pollution species against coronavirus and mask wearing very vague based on the GAM analysis.
The quality of presentation is not up to the standards for publication in this journal at this point. The entire manuscript needs extensive revision, rewriting, and reorganization. Thus, I would not recommend publishing this manuscript. Nevertheless, the data are of value, and an improved manuscript should be publishable in the future. I recommend the manuscript should be professionally checked by the English editing services. The above are the major issues:
Major comments:
(1) In the abstract (line 09-21), the authors fail to concisely deliver the key findings of the manuscript. In line 14-20, I found it difficult to understand how the results can be related to wearing mask.
(2) In the introduction (line 26-81), this section is not organized at all. The authors fail to outline the issues, what has been studied in the literature before and how the present manuscript can address those issues. In addition, the sections in line 32-54 are too long without any citations. In line 69-72, I am not confident about information provided from a mobile instant messenger.
(3) In Section 2.1-2.4 (line 82-183), there are approximately less than 50% of the city’s data (159 out of 339 cities) used for this study. How can the authors warrant the results from the GAM analysis are representative? In line 124, there is lack of citation for the dataset obtained from Beijing Standard Time. In line 148-149, the idea of introducing high-speed rail line numbers as a factor variable is unclear. The people who infected with COVID-19 could be taking other types of transport and hence spread the virus? How can the authors ensure the input of factor variable (high-speed rail line numbers) reflect the actual condition?
(4) In Section 3 (line 184-283), in line 230-233, please provide citations to support your claims. In line 253-254, can you please provide further supporting information why there was no reported cases when the average PM2.5 AQI level was right or hazardous during the study period.
(5) In Section 4 (line 284-362), in line 285-291, there is no discussion about your findings. In line 292-312, a lot of texts have been devoted to previous studies without explanations about why are they related to your results. In line 319-335, the section is too long without getting to the main points. In line 342-343, I am very concerned about the accuracy of the provided information.
(6) In Section 5 (line 363-374), please revise your conclusion and avoid using term such as “Nth” as the authors have not specify the definition. How did your key findings relate to your conclusion?
Minor comments:
1. Keywords, line 22 and 23
Please use other keywords to replace “ecological study” and “social distancing”.
- Section 1, line 61
Please use alternative to replace “rate of mask wearing”.
- Section 2.1, Figure 1, line 91
The map quality of China is very poor. Please use appropriate scale.
- Section 2.2, line 131-132
Please re-arrange the sentence to another section.
- Section 2.2, line 138-139
Please use >301 and >250.5 instead.
- Section 3, Table 1, line 194-202
Please provide footnotes to explain why there are only 141 and 158 cites for the O3 and population density respectively.
- Section 3, Table 1, line 202
Please use ozone instead of trioxygen.
- Section 3, Figure 4-5, line 244-248 and 249-252
Please properly align your figures.
- Section 4, line 298
Please subscript “3”.
- Section 4, line 319 and 321
Please revise “virgin soil epidemic” and “slowed”.
Author Response
I write you in regard to manuscript ijerph-1062570 entitled "Population response to air pollution and the risk of coronavirus disease in Chinese cities during the early pandemic period” which you submitted to International Journal of Environmental Research and Public Health. This study targets the effect of health behaviors against air pollution on reducing the risk of COVID-19 during the initial pandemic. The attack rate was further compared with the level of air pollution using a generalized linear model after adjusting for confounders. The study also evaluated the health behaviors towards using face mask to prevent infections.
2-1. This study aims to add knowledge about air pollution and the risk of coronavirus in different areas in China. However, the authors seem to be too ambitious using a rather small dataset (2 weeks) to predict the abovementioned association.
Response:
This study is an ecologic study, and specifically designed to demonstrate the health behavior of the population exposed to COVID-19 infection, while protective behavior including mask wearing was less prominent. By selecting the window period of early pandemic period, we intended to demonstrate the relationship between potential level of mask wearing while people were less aware of COVID-19 pandemic and COVID-19 incidence. This is key design of this study and illustrated why only initial two weeks cases were selected as a study period.
As the reviewer indicated, the number of data in this study was small, but we fitted using the nonlinear model (GAM) suitable for predicting association and direction. Our actual sample size to be analyzed was 6821 cases obtained daily from each area during the study period of 2 weeks.
In the early COVID-19 pandemic, the population has worn a mask for reasons air pollution in China. It prevented the COVID-19. When COVID-19 was first reported, people were unaware that COVID-19 was an infection disease. This study is based on the time before the Chinese government intervened.
- To estimate the case of COVID-19 that occurred before the Chinese government locked down cities in Wuhan and Hubei, reflected onset-diagnosis time 13 days as of January 23, 2020, when Wuhan was locked down. Thus, based on cases that occurred until February 5, 2020.
- The first case other than the Hubei was reported in China on January 21, 2020. We reflected onset-diagnosis time 13 days as of January 21. Hence, we used air pollution data for 16 days, started January 8, 2020, until January 23, 2020, when Wuhan was locked down.
- The case (from January 21, 2020 to February 5, 2020) and pollution levels (from January 8, 2020 to January 23, 2020) are not day-to-day matchings.
- The cumulative case (until February 5, 2020) and the average air pollution level.
Hence, we decided the data set is suitable.
- I am particularly unsure about the data representativeness for the generalized additive model (GAM) outcome. I also found the links between several air pollution species against coronavirus and mask wearing very vague based on the GAM analysis.
Response:
As a result of GAM analysis, PM10, O3, CO, and SO2 were not statistically significant, and PM2.5 and NO2 were statistically significant. We think it is worth attention to why higher PM2.5 and NO2 decreased the initial risk of COVID-19, not other air pollutants.
We verified the effects of PM2.5 and NO2 among air pollutants on health behavior. The website (https://aqicn.org) provides air quality based on PM2.5 of the real-time air pollution index (Figure 2). When searching for “Real-time air pollution” on Baidu* (https://www.baidu.com/), China's largest web portal, the air quality search site in the top line (http://www.air-level.com/) also provides air pollution based on PM2.5 (Figure R21). Thus, PM2.5 AQI based when China's population promotion health behavior against air pollution. The mask purchase index increased as the air pollution AQI level increased (Table R21) [1].
Figure R21. Left: The real time air pollution in Wuhan, China. Source: https://aqicn.org cited 2021 Jan 22, Right: The real time air pollution in China Source: http://www.air-level.com/ cited 2021 Jan 22 (for figures and tables, see attached file).
Table R21. The mask purchase index and AQI and mask index. Source: Zhang and Mu
The mean annoyance scores at population level and mean PM2.5 and NO2 concentrations correlations were high (Figure R22) [2]. As the concentration of PM2.5 increased, the number of Anti PM2.5 masks Search frequency increased (Figure R23) [3]. Hence, PM2.5 and NO2 can be factors promoting health behavior in the population.
The only significant result of PM2.5 and NO2 in the GAM model is strong evidence of this study's hypothesis that the concentration of PM2.5 and NO2 promoted the population's health behavior, thereby increasing the wearing rate of masks.
.
Figure R22. Left: PM2.5 and annoyance scores, Right: NO2 and annoyance scores <Source: Rotko et al. >
Figure R23. PM2.5 and search frequency of Anti-PM mask <Source: Liu et al. >
The quality of presentation is not up to the standards for publication in this journal at this point. The entire manuscript needs extensive revision, rewriting, and reorganization. Thus, I would not recommend publishing this manuscript. Nevertheless, the data are of value, and an improved manuscript should be publishable in the future. I recommend the manuscript should be professionally checked by the English editing services. The above are the major issues:
Major comments:
(1) In the abstract (line 09-21), the authors fail to concisely deliver the key findings of the manuscript. In line 14-20, I found it difficult to understand how the results can be related to wearing mask.
Response:
Thank you for the suggestion. We have revised the abstract (L14-23).
(2) In the introduction (line 26-81), this section is not organized at all. The authors fail to outline the issues, what has been studied in the literature before and how the present manuscript can address those issues. In addition, the sections in line 32-54 are too long without any citations.
Response:
Thank you for the suggestion. We have revised the abstract (L42-79).
(3) In line 69-72, I am not confident about information provided from a mobile instant messenger
Response:
We cited a study that revealed that it is related to seeking health information, which is the act of searching Haze-related information in mobile instant messengers, rather than the reliability of the information provided by mobile instant messengers. Several citations have been added to support the fact that air pollution is a cause for health-seeking behavior (In line 74-78).
(3.1) In Section 2.1-2.4 (line 82-183), there are approximately less than 50% of the city’s data (159 out of 339 cities) used for this study. How can the authors warrant the results from the GAM analysis are representative?
Response:
We excluded cities without air quality measurements, with <10 cases reported during the study period, and all cities (17 cities) in the Hubei province, including Wuhan. We included all Chinese cities that meets inclusion criteria and we believe that 159 cities are appropriate number to represent China at the time of the study, the study period is early pandemic COVID-19. Also, GAM is a for inferring the dose-response relationship of health effect when non-linear relationship is expected and it is a flexible model to represent the non-linear or linear relationship between two variables. To estimate the parameters, we have applied generalized linear model (GLM), which is widely applied in the environmental epidemiology. Excluding cities with fewer occurrences increases the validity of the analysis to identify disease trends.
(3.2) In line 124, there is lack of citation for the dataset obtained from Beijing Standard Time.
Response:
We citated DXY·DY Doctor. COVID-19 Global Pandemic Real-time Report [reference 17]. It is stated that Beijing Standard Time was used.
(3.3) In line 148-149, the idea of introducing high-speed rail line numbers as a factor variable is unclear. The people who infected with COVID-19 could be taking other types of transport and hence spread the virus? How can the authors ensure the input of factor variable (high-speed rail line numbers) reflect the actual condition?
Response:
The spread of infections could be associated with the domestic transportations in mainland China [4]. It was not related to other means of transportation except railroads in the early COVID-19 outbreak in China. (Figure R24) [4].
Figure R24. Railway's impact on the spread of infection in mainland China. Source : Zhao et al.
Table R22. The association between transportation and number of imported cases Source : Zhao et al.
The regions connected by express trains to Wuhan – such as Shenzhen, Shanghai and Beijing – had five, two and two confirmed cases, respectively on 21 January, the early stage of COVID-19 in China [5].
Hence, factor variable (high-speed rail line numbers) reflect the actual condition.
(4.1) In Section 3 (line 184-283), in line 230-233, please provide citations to support your claims.
The larger the population is, the higher the transportation volumes, economic activity, air pollution, and virus infections (Figure R25) [6]. population density and wind explain 94% of the variance in COVID-19 propagation [7].
Figure R25. Interactions between anthropogenic factors, natural factors, and virus infections. [a]
(4.2) In line 253-254, can you please provide further supporting information why there was no reported cases when the average PM2.5 AQI level was right or hazardous during the study period.
Response:
Thank you for the suggestion. We have completely revised the result to clearly demonstrate the relationship expressed as AQI and risk of the infection (L252-261).
(5.1) In Section 4 (line 284-362), in line 285-291, there is no discussion about your findings.
Response: We have revised the discussion to include this suggestion (L295-318).
(5.2) In line 292-312, a lot of texts have been devoted to previous studies without explanations about why are they related to your results.
Response: In line 318 : It has changed in the revised text. In line 296-312: It has changed in the revised text.
We provided the results of previous studies on the negative effects of air pollution on COVID-19 and the positive effects of COVID-19 on air pollution. These previous studies are about the direct relationship between air pollution and COVID-19. Our study is different from previous studies by using air pollution as a surrogate indicator for the health behavior of wearing a mask. We rearranged the manuscript with the help of your opinion.
(5.3) In line 319-335, the section is too long without getting to the main points.
Response: We have resived the paragraph.
(5.4) In line 342-343, I am very concerned about the accuracy of the provided information.
Response:
People respond to higher levels of air pollution by buying more masks in China. They respond to both government's pollution alerts (determined by PM2.5 exceeding key thresholds) and to the level of outdoor PM2.5[8]. the daily sales of masks on the days when the government has issued a “heavily polluted” and “severely polluted” alert are 2.9 and 7.2 times those during a “blue sky” day (Figure 11) [8]. During a hazy week at the end of 2013, mask sales reached 760,000[8]. Hence, wearing a mask is usually a way to prepare for air pollution.
Table R23. Daily Internet sales of self-protection products as a function of air pollution.
(6) In Section 5 (line 363-374), please revise your conclusion and avoid using term such as “Nth” as the authors have not specify the definition. How did your key findings relate to your conclusion?
Response: Our key finding was that wearing a mask caused by air pollution reduced the risk of COVID-19. It is the basis for the importance of wearing a mask.
In line 396-401: It has changed in the revised text.
Minor comments:
(1) Keywords, line 22 and 23: Please use other keywords to replace “ecological study” and “social distancing”.
Response: We have revised the keywords.
(2) Section 1, line 61: Please use alternative to replace “rate of mask wearing”.
Response: As the reviewer indicated, “rate of mask wearing” has changed to “mask-wearing rate”.
(3) Section 2.1, Figure 1, line 91: The map quality of China is very poor. Please use appropriate scale.
Response: Figure 1 has newly represented in the revised manuscript. Resolution of the figure has been improved.
(4) Section 2.2, line 131-132: Please re-arrange the sentence to another section.
Response: it has been re-arranged.
(5) Section 2.2, line 138-139: Please use >301 and >250.5 instead.
Response: Line 138-139 has changed in the revised text.
(6) Section 3, Table 1, line 194-202: Please provide footnotes to explain why there are only 141 and 158 cites for the O3 and population density respectively.
Response: I added a description to the footnote (In line 220).
(7) Section 3, Table 1, line 202: Please use ozone instead of trioxygen.
Response: Table 1 has changed in the revised text.
(8) Section 3, Figure 4-5, line 244-248 and 249-252: Please properly align your figures.
Response: We have revised the figures accordingly.
(9) Section 4, line 298: Please subscript “3”.
Response: It has been revised.
(10) Section 4, line 319 and 321: Please revise “virgin soil epidemic” and “slowed”.
Response: As the reviewer indicated, “virgin soil epidemic” has changed to “An emerging infectious diseases” (In line: 319-320). As suggested, “slowed” has changed to “decreased” (In line: 321).
References
- Zhang, J.; Mu, Q. Air pollution and defensive expenditures: evidence from particulate-filtering facemasks. J. Environ. Econ. Manag. 2018, 92, 517-536.
- Rotko, T.; Oglesby, L.; Künzli, N.; Carrer, P.; Nieuwenhuijsen, M.J.; Jantunen, M. Determinants of perceived air pollution annoyance and association between annoyance scores and air pollution (PM2.5, NO2) concentrations in the European EXPOLIS study. Atmos. Environ. 2002, 36(29), 4593-4602
- Liu, T.; He, G.; Lau, A. Avoidance behavior against air pollution: evidence from online search indices for anti-PM 2.5 masks and air filters in Chinese cities. Environ. Econom. Policy Stud. 2018, 20(2), 325-363.
- Zhao, S.; Zhuang, Z.; Ran, J.; Lin, J.; Yang, G.; Yang, L.; He, D. The association between domestic train transportation and novel coronavirus (2019-nCoV) outbreak in China from 2019 to 2020: A data-driven correlational report. Travel Med. Infect. Dis. 2020, 33, 101568.
- Kang, D.; Choi, H.; Kim, J.H.; Choi, J. Spatial epidemic dynamics of the COVID-19 outbreak in China. Int. J. Infect. Dis. 2020, 94, 96-102.
- Copiello, S.; Grillenzoni, C; The spread of 2019-nCoV in China was primarily driven by population density. Comment on “Association between short-term exposure to air pollution and COVID-19 infection: Evidence from China” by Zhu et al.. Sci Total Environ 2020 20, 744,141028
- CoÅŸkun, H; Yıldırım, N; Gündüzc, S. The spread of COVID-19 virus through population density and wind in Turkey cities. Sci. Total Environ. 2021,751,
8.Sun, C.; Kahn, M.E.; Zheng S. Self-protection investment exacerbates air pollution exposure inequality in urban China. Ecol. Econ. 2017, 131, 468-474
- Ban, J.; Zhou, L.; Zhang, Y.; Anderson, G.B.; Li, T. The health policy implications of individual adaptive behavior responses to smog pollution in urban China. Environ. Int.2017, 106, 144–152.
- Zhang, J.; Mu, Q. Air pollution and defensive expenditures: evidence from particulate-filtering facemasks. J. Environ. Econ. Manag. 2018, 92, 517-536.
- Center for disease control and prevention (CDC). Personal Protective Equipment: Questions and Answers. Available online: https://www.cdc.gov/coronavirus/2019-ncov/hcp/respirator-use-faq.html (accessed on 25 January 2021).
Baidu*: Ranks among the top in the world by Alexa.com traffic metrics (cited 2021 Jan 22. https://www.alexa.com/topsites)

Reviewer 3 Report
The manuscript entitled “Population response to air pollution and the risk of coronavirus disease in Chinese cities during the early pandemic period” aimed to highlight the importance of wearing a mask to prevent infection in a situation when vaccines and therapeutics are widely used and the population has not entered herd immunity. This is an important work and I appreciate the authors’ effort, but I still have some questions about the paper.
- Introduction is inappropriately written, the descriptions of the significance of this study were relatively poor, please enrich it.
- Line 55: The description is wrong. There is still no international conclusion about the country where COVID-19 first occurred. China is the first country to report COVID-19.
- Figure 1: It is recommended to enlarge Fig. 1 appropriately and modify the sharpness of the image. And I suggest the authors change the color scheme used in Fig. 1 to make the diagram more readable.
- Line 139-140: Authors assumed that the population wore masks on days when the AQI exceeded 150, the level that affects the general population's health. Please provide scientific research or deductive evidence for the hypothesis.
- In the discussion, it is important to note that most masks worn to prevent air pollution do not reduce the risk of COVID-19.
- The reference format should be consistent, for example, the journal name should be written in full or abbreviated, some references require additional volume number and issue number.
- In the manuscript, the practical significance of the elaboration is not profound enough, the authors need to elaborate in more detail on the significance of this work in the conclusions. Using air pollution data to promote healthy behaviour is not very feasible.
Author Response
The manuscript entitled “Population response to air pollution and the risk of coronavirus disease in Chinese cities during the early pandemic period” aimed to highlight the importance of wearing a mask to prevent infection in a situation when vaccines and therapeutics are widely used and the population has not entered herd immunity. This is an important work and I appreciate the authors’ effort, but I still have some questions about the paper.
Author’s response: We deeply appreciate your comprehensive review and we have revised and commented in response to your comments accordingly.
(3-1) Introduction is inappropriately written, the descriptions of the significance of this study were relatively poor, please enrich it.
Response: We have revised the introduction to strengthen the significance of this study (L69-78).
(3-2) Line 55: The description is wrong. There is still no international conclusion about the country where COVID-19 first occurred. China is the first country to report COVID-19.
Response: We have revised the description into the first report basis (L30, 49, 57).
(3-3) Figure 1: It is recommended to enlarge Fig. 1 appropriately and modify the sharpness of the image. And I suggest the authors change the color scheme used in Fig. 1 to make the diagram more readable.
Response: Figure 1 has been modified in the revised manuscript. Figure resolution has been improved.
(3-4) Line 139-140: Authors assumed that the population wore masks on days when the AQI exceeded 150, the level that affects the general population's health. Please provide scientific research or deductive evidence for the hypothesis.
Response: According a questionnaire survey in China, Most respondents (76.4%) increased their use of anti-PM masks when outdoors during smog events (Table R31) [9]. There is a positive correlation between PM2.5 and the anti-PM mask online search index (Figure R31) [3]. When PM2.5 increases by 10 µg/m3, daily online orders of anti-PM2.5 masks increase by 7166 pieces [3]. Zhang and Mu estimated that 100-point increase in the AQI increases the total purchase of anti-PM2.5 facemasks and all facemasks by 70.6% and 50.4%, respectively [10]. Thus, we assumed that when the concentration of PM2.5 was increased, the mask was used as the health behavior. It is the reference on the AQI 150, which affects the health of the general public.
Table R31. Percentages of different behavioral changes about hazardous air pollution (for figures and tables. see attached file).
Figure R31. Scattered plot of online search indices and PM2.5 in Beijing, The logarithms of online search indices for anti-PM2.5 masks plotted on the y axis, and daily PM2.5 is plotted on the x axis
(3-5) the discussion, it is important to note that most masks worn to prevent air pollution do not reduce the risk of COVID-19.
Response: We agree that anti-PM mask is not developed for the prevention of virus transmission. However, we have observed that according to the logical conclusion of our study, anti-PM2.5 mask, worn to protect against the hazard of air pollution, may also work to prevent risk of COVID-19 infection. Theoretically, US Center for disease control and prevention (CDC) reported that N95 respirators reduce the wearer’s exposure to airborne particles, from small particle aerosols to large droplets [12]. N95 respirators are tight-fitting respirators that filter out at least 95% of particles in the air, including large and small particles [11]. They are guiding public to wear N95 PPE to prevent COVID-19 [11]. Even though it may not perfect in protecting from the virus particle-included aerosols and droplets, in practice, it may work to reduce transmission.
(3-6) The reference format should be consistent, for example, the journal name should be written in full or abbreviated, some references require additional volume number and issue number.
Response: Reference format has been reformatted in the revised text (L392-472).
(3-7) In the manuscript, the practical significance of the elaboration is not profound enough, the authors need to elaborate in more detail on the significance of this work in the conclusions. Using air pollution data to promote healthy behaviour is not very feasible.
Response:
It has changed in the revised text (L350-353, 377-378) .
References
- Zhang, J.; Mu, Q. Air pollution and defensive expenditures: evidence from particulate-filtering facemasks. J. Environ. Econ. Manag. 2018, 92, 517-536.
- Rotko, T.; Oglesby, L.; Künzli, N.; Carrer, P.; Nieuwenhuijsen, M.J.; Jantunen, M. Determinants of perceived air pollution annoyance and association between annoyance scores and air pollution (PM2.5, NO2) concentrations in the European EXPOLIS study. Atmos. Environ. 2002, 36(29), 4593-4602
- Liu, T.; He, G.; Lau, A. Avoidance behavior against air pollution: evidence from online search indices for anti-PM 2.5 masks and air filters in Chinese cities. Environ. Econom. Policy Stud. 2018, 20(2), 325-363.
- Zhao, S.; Zhuang, Z.; Ran, J.; Lin, J.; Yang, G.; Yang, L.; He, D. The association between domestic train transportation and novel coronavirus (2019-nCoV) outbreak in China from 2019 to 2020: A data-driven correlational report. Travel Med. Infect. Dis. 2020, 33, 101568.
- Kang, D.; Choi, H.; Kim, J.H.; Choi, J. Spatial epidemic dynamics of the COVID-19 outbreak in China. Int. J. Infect. Dis. 2020, 94, 96-102.
- Copiello, S.; Grillenzoni, C; The spread of 2019-nCoV in China was primarily driven by population density. Comment on “Association between short-term exposure to air pollution and COVID-19 infection: Evidence from China” by Zhu et al.. Sci Total Environ 2020 20, 744,141028
- CoÅŸkun, H; Yıldırım, N; Gündüzc, S. The spread of COVID-19 virus through population density and wind in Turkey cities. Sci. Total Environ. 2021,751,
8.Sun, C.; Kahn, M.E.; Zheng S. Self-protection investment exacerbates air pollution exposure inequality in urban China. Ecol. Econ. 2017, 131, 468-474
- Ban, J.; Zhou, L.; Zhang, Y.; Anderson, G.B.; Li, T. The health policy implications of individual adaptive behavior responses to smog pollution in urban China. Environ. Int.2017, 106, 144–152.
- Zhang, J.; Mu, Q. Air pollution and defensive expenditures: evidence from particulate-filtering facemasks. J. Environ. Econ. Manag. 2018, 92, 517-536.
- Center for disease control and prevention (CDC). Personal Protective Equipment: Questions and Answers. Available online: https://www.cdc.gov/coronavirus/2019-ncov/hcp/respirator-use-faq.html (accessed on 25 January 2021).
Baidu*: Ranks among the top in the world by Alexa.com traffic metrics (cited 2021 Jan 22. https://www.alexa.com/topsites)

Round 2
Reviewer 1 Report
The authors should better and more clearly explain in the abstract before the final conclusions (those ones concerning face masks) thei detection of a kind of "paradox" effect (as they well present in the discussion), so that in the most polluted cities it has been observerd lower incidence of COVID, probably due to the more effective community-based measure and individual protections (wearing face masks) implemented routinely in the most polluted cities.
If the author better clarify in the abstract and conlcusions this concept (speaking about an "apparently paradox effect") there are no more obstacles for the publication
Author Response
Thank you for the suggestion. We have revised the abstract (L20-25) and conclusion (L366-376) section to make the point clearer.
Reviewer 2 Report
(1) In the abstract (line 20-22), please revise the statement to make it more concise why negative association between air pollution level and attack rate could reduce the risk of COVID-19 during the period.
(2) In the conclusions (line 365-366), please delete the sentence “The Nth Wave of COVID-19 infection continues worldwide.”
Author Response

(The authors gave the same response as above.)

Reviewer 3 Report
I think the authors have fully addressed my comments and the quality of this paper has been improved, so I recommend the publication of this paper.
Author Response

(The authors gave the same response as above.)
